# Integrated Waveform Design Based on UAV MIMO Joint Radar Communication

Hao Ma , Jun Wang *, Xin Sun and Wenxin Jin

School of Electronic and Information Engineering, Beijing Jiaotong University, Beijing 100044, China; 16111020@bjtu.edu.cn (H.M.); xsun@bjtu.edu.cn (X.S.); 17111022@bjtu.edu.cn (W.J.)
* Correspondence: wangjun1@bjtu.edu.cn; Tel.: +86-138-1017-2792

**Abstract:** The problem of orthogonal waveform construction in multiple input/multiple output (MIMO) radar communication integration greatly limits the realization of integration technology. In the unmanned aerial vehicle (UAV) MIMO antenna scenario, an orthogonal integrated waveform suitable for a MIMO antenna is designed using a sub−LFM−BPSK waveform combined with a chaotic spread spectrum code. After spread spectrum processing, each MIMO antenna transmits different communication data for orthogonal spread spectrum processing, which is suitable for the omnidirectional detection of MIMO application scenarios; moreover, the closed-form expressions of the integrated orthogonal waveform under certain constraints are derived. Finally, the simulation proves that the integrated orthogonal waveform set in the UAV MIMO scenario has excellent radar detection and communication capabilities.

**Keywords:** orthogonal waveform; radar communication integration; MIMO antenna; spread spectrum processing



## 1. Introduction

With the development of 5G communication technology, the number of wireless terminals has skyrocketed, resulting in increasing conflicts between wireless communication and radar, and the frequency interference between them disrupts their coexistence and development [1,2]. The development of unmanned aerial vehicles (UAVs) relies on the new generation of information technology; as important parts of UAVs, communication and radar are indispensable functions. Antenna configuration, spectrum interference and power consumption have a significant impact on UAV performance. In this case, the emergence of joint radar communication technology effectively solves the interference and spectrum conflict between the two and greatly improves the performance of the UAV [3–5]. As an integration technology, radar communication integration is mainly divided into two main technology routes: spectrum sharing and complete integration [6]. In spectrum-sharing technology, the radar and communication each use existing hardware equipment, but pre-coding and other technologies allow the functions to coexist within a segment of the spectrum, and mutual interference is minimized. In complete integration technologies, the system uses a set comprising a hardware platform and integrated signal to achieve radar and communication functions at the same time in one segment of the spectrum, which is a development of radar communication integration technology. The complete integrated technology reduces the platform's power consumption, antenna volume and energy consumption; lowers technical costs; eliminates dual-function interference; and effectively promotes the integration of multiple electronic platforms. Radar communication integration has been widely studied by the academic and industrial communities, and its application field has gradually expanded from military use to include rescue, intelligent transportation, UAV control and other civil fields. In the field of UAV control, considering the requirements of UAVs in terms of volume, energy consumption and detection qual-

ity, the complete integrated technology is one of the key technologies urgently needed by UAVs.

At present, multiple-input/multiple-output (MIMO) antenna technology has gradually become mainstream in the communication field, and it has also been gradually applied to the field of UAVs. Inspired by this, scholars in the field of radar have proposed MIMO radar technology. Compared with single-antenna radar, MIMO radars have excellent narrow beamforming and better freedom, which greatly improves their radar detection ability [6]. The spectrum sharing of MIMO communication and MIMO radar has gradually developed into the complete integration of MIMO radar communication and realized the integration of the two. The waveform design has also evolved from the integrated waveform for a single antenna to the integrated waveform for a MIMO antenna.

Over the past few decades, the academic and industrial communities have extensively studied integrated waveform design based on a single antenna, leading to a number of achievements. The authors of [7] proposed the LFM−MSK integrated signal by combining the LFM radar signal with MSK communication, and they analyzed the ambiguous function characteristics of the integrated signal. The authors of [8] further analyzed the time–frequency characteristics of LFM−MSK and proposed a scheme to ensure high-throughput data transmission under the condition of spectrum without diffusion. In [9], an Oppermann multiphase sequence spread spectrum code was proposed to realize the distinction between radar and communication. The authors of [10] proposed an LFM−CPM integrated waveform design and analyzed the waveform's ambiguous function with the use of data reconstruction to improve spectrum performance and its application in the field of intelligent transportation. The authors of [11–13] provided detailed design and performance analyses of radar communication integration based on OFDM.

There have also been many technological achievements in the field of MIMO radar communication integration. In [14], Amin et al. first proposed that the side-lobe level of the MIMO radar transmit beam was used to realize the communication function, and the communication receiver used the energy detection algorithm to realize the communication demodulation. In [15], Amin et al. also proposed an integrated method to transmit communication data within a pulse by using the side-lobe level and phase of MIMO waveforms. In [16], a convex optimization method was used to design an integrated transmission direction graph that can improve communication performance and radar detection function under the condition of minimizing multiple interferences by communication users. In [17–20], Liu Fan et al. undertook detailed studies of the performance of MIMO radar communication integration in spectrum sharing, beamforming, waveform design scheme and vehicle-to-vehicle application scenarios.

By analyzing the above single-antenna integrated waveform scheme and MIMO radar communication integration scheme, it can be observed that there are some problems in the existing technical methods. 1. When the MIMO integrated system is used to achieve radar and communication at the same time, the beam direction formalization design is mainly considered, whereas the specific integrated waveform expression is rarely considered, and the universality of the integrated waveform in the field of radar communication integration is not considered. 2. From the perspective of analysis convenience, some studies [17–20] directly accept the communication signal with constellation distribution as the integrated waveform when constructing the MIMO integrated waveform system. Although it is easier to analyze the communication performance of the entire integrated system from a communication perspective, its performance is not comprehensively considered in terms of radar application scenarios. 3. Although the single-antenna integrated waveform design technology is mature, when applied to MIMO radar communication integration, it is difficult to ensure that each antenna transmits an orthogonal waveform to construct omnidirectional detection signals. The orthogonal waveform problem limits the specific application of integrated MIMO waveform design.

Based on the above problems, this paper proposes an integrated omnidirectional orthogonal waveform method for MIMO radar communication in a UAV scenario, and the application scenario diagram is shown in Figure 1.

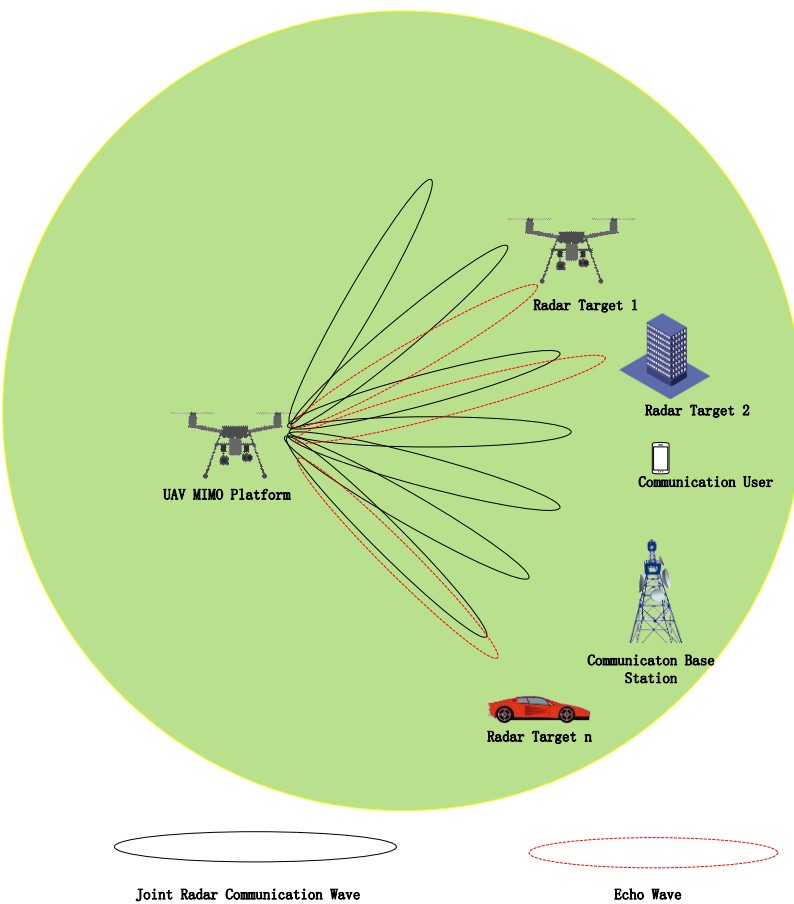

**Figure 1.** UAV-based MIMO radar communication integration application scenario.

The contributions of this paper are as follows:

(1) Based on the integrated waveform of a single antenna [21], the integrated orthogonal waveform set in the MIMO scenario is constructed by adopting the orthogonal spreading code method.

(2) The radar detection performance and the communication performance of the orthogonal waveforms of the integrated MIMO system in the UAV scenario are analyzed, and the mathematical expression is derived.

(3) The performance of the orthogonal waveforms is simulated, and the results confirm that the integrated system has excellent radar and communication performance.

The joint radar communication system can realize the mutual benefits of communication and radar and can be used in a variety of scenarios in the future. In the vehicles-to-vehicle scenario, it is necessary to recognize the environment; identify the position, speed and direction of movement of vehicles; and detect abnormal events on the road. The integrated radar communication system can detect the state of traffic flow on the road in real time. It can achieve the efficient coordination of people, vehicles and roads; ensure traffic safety; and improve the operational efficiency of the traffic system. In the smart home, joint radar communication technology can use a Wi-Fi wireless signal to detect human movements and behavior and provide more information for smart home systems.

The structure of this paper is as follows. The first part analyzes the challenging problem of the orthogonal waveform set in the integrated waveform MIMO UAV scenario. The second part proposes the orthogonal integrated waveform model applied to the UAV scenario. The third part analyzes the radar performance and communication performance

of the integrated waveform. In the fourth part, the ambiguity function, the azimuth–range map and the spectral performance of the MIMO waveform are verified via experimental simulation. The fifth part gives the conclusion.

Notation: Lower case (i.e., $a$) and upper case (i.e., $A$) letters denote complex vectors and matrices, respectively. $\mathbb{C}$ represents the set of complex numbers. $[\cdot]^c$, $[\cdot]^T$ and $[\cdot]^H$ denote the conjugate, transpose and conjugate–transpose operation, respectively. $\|\cdot\|_F$ denotes the Frobenius norm; $tr(\cdot)$ denotes the diagonal operation. $\text{Re}\{\cdot\}$ is the real part of the argument.

## 2. System Model

Common MIMO radar systems are divided into co-location MIMO radar and distributed MIMO radar. In co-location MIMO radar systems, the transceiver antennas are centralized, and the targets are distributed in the far field. In a distributed MIMO radar system, the transceiver antennas are distributed throughout the coverage area. In the UAV radar communication integration scheme, the MIMO antenna layout is co-location; i.e., a single UAV is equipped with both the transmit and receive arrays. The detection target is located in the far field as a point target, and the transmit and receive angles of the target match those of the UAV's transceiver antenna array. The downlink communication user is in the far field. The detection target and the downlink communication user are in the line of sight.

Different scenarios should be considered when the UAV with a MIMO antenna uses the integrated waveform to perform tasks. When the UAV is in the detection scenario, each antenna must transmit orthogonal waveforms to form an omnidirectional beam that can realize the communication function while realizing target detection. When the UAV is in the target-tracking scenario, it must transmit a partially orthogonal waveform to form a directional beam and realize the communication task while tracking the target. In the target-tracking scenario, the UAV's waveform design is similar to the integrated waveform design of phased-array-based radar communication integration. This paper focuses on the design of orthogonal waveforms for omnidirectional detection.

In the MIMO antenna scenario, orthogonal integrated waveforms are mainly divided into three technologies: code orthogonality, time-division orthogonality and frequency division orthogonality.

In study [2], a time-division orthogonal waveform was proposed. In the integrated MIMO system, each antenna transmits the integrated waveform signal in different time slots. The working process is as follows: First, the first antenna completes the transmission of the integrated waveform and receives the echo; second, the second antenna completes the transmission of the integrated waveform and receives the echo, and so on. The waveforms between the antennas are orthogonal because different antennas perform detection operations in different time slots. An orthogonal waveform set is designed for the MIMO antenna scenario, which is suitable for the omnidirectional detection working scenario. Although this scheme is easy to operate, the time resources allocated to each antenna are limited, and the space–time resources cannot be fully utilized. As the number of antennas increases, there are problems comprising limited communication rate and limited range.

In study [5], the LFM wave is modulated onto several equally spaced carrier frequencies to form an orthogonal frequency signal set used in the integrated MIMO system. This is equivalent to using LFM waveforms in OFDM subcarriers to replace conventional communication data. This type of technology has the disadvantage of a high peak-to-average power ratio. In addition, radar detection performance is reduced when multiple antennas are used.

The technology used in this paper is orthogonal coding technology. This technology uses the spread spectrum code to assemble the sub$-$LFM$-$BPSK of a single antenna to generate an orthogonal set of waveforms. A large number of orthogonal code sequences can be generated by chaotic spreading codes. Based on these sequences, a large number of orthogonal integrated waveform sets can be generated, which can improve the communica-

tion rate and is also suitable for MIMO antenna scenarios. The performance comparison of the three orthogonal waveform sets is shown in Table 1.

**Table 1.** Comparison of three orthogonal waveform methods.

| Orthogonal Waveform Method | Method in This Paper | Literature [2] | Literature [5] |
|---|---|---|---|
| Waveform characteristics | Mutually orthogonal coded waveforms | No specific requirements | Orthogonal carrier frequencies |
| Key technologies | Spread code orthogonality | The design of divided time slots | Carrier orthogonality |
| Advantages | 1. High communication rate; 2. Suitable for large-scale MIMO antenna occasions. | 1. Strong waveform applicability; 2. Simple hardware circuit. | 1. High spectrum utilization; 2. High communication rate. |
| Disadvantages | 1. Complex hardware circuit | 1. Low communication rate; 2. Short detection distance. | 1. Complex hardware circuit; 2. High peak-to-average power ratio. |

### 2.1. Integrated Omnidirectional Waveform Design of UAV

To ensure that the power amplifier of the integrated system is in the saturated region, the transmit signal must have a constant envelope. The LFM waveform, with a constant envelope that matches the operating environment of radar power amplification, is the most comprehensive radar waveform available. The integrated signal uses the LFM radar wave as the original signal to ensure the stability of the existing radar signal processing system, which has excellent performance advantages. Based on the LFM radar waveform embedded with the communication data, the integrated waveform is realized in this paper.

In the scenario in which a MIMO antenna array is integrated into a UAV, assuming that the number of transmit antennas and receive antennas of the integrated terminal of the UAV are $N_t$ and $N_r$ respectively, the direction vectors of the waveform transmitting array and receiving array of the UAV are, respectively expressed as follows [18]:

$$a(\varphi) = \begin{bmatrix} 1 & \exp(j\alpha d_t \sin \varphi) \cdots & \exp(j\alpha d_t (N_t - 1) \sin \varphi) \end{bmatrix}^T \in \mathbb{C}^{N_t \times 1} \tag{1}$$

$$b(\varphi) = \begin{bmatrix} 1 & \exp(j\alpha d_r \sin \varphi) \cdots & \exp(j\alpha d_r (N_r - 1) \sin \varphi) \end{bmatrix}^T \in \mathbb{C}^{N_r \times 1} \tag{2}$$

In Equations (1) and (2), $\varphi$ is the azimuth of the target, $\alpha$ is the wave number, and $d_t$ and $d_r$ represent the distance intervals of the transmitting antenna array and the receiving antenna array, respectively.

As an integrated system evolves from a single antenna to a MIMO system, the orthogonality of the waveforms transmitted by each antenna is an important issue to consider. In the existing single-antenna integrated waveform, the waveform has adopted orthogonal processing, which can effectively solve the problem of MIMO radar communication integration detection.

The performance of the single-antenna integrated waveform proposed in [21] was analyzed according to its waveform characteristics, although LFM−BPSK could realize the communication function, its radar detection function deteriorated sharply. The performance analysis shows that the performance degradation of the LFM−BPSK scheme is caused by the large phase shift. Thus, the sub−LFM−BPSK scheme is proposed; therein, the phase shift of the data for the integrated waveform is no longer $\pi/2$ or $-\pi/2$ but $\theta$ or $-\theta(\theta < \pi/2)$. The sub−LFM−BPSK method not only reduces communication performance but also reduces radar detection loss.

To compensate for the loss in communication performance of the integrated waveform, communication data are processed using a chaotic spread spectrum code. The integrated signal is generated by a sub−LFM−BPSK communication signal via spread spectrum pro-

cessing. The expressions of the sub$-$LFM$-$BPSK modulation signal and spread spectrum omnidirectional integrated waveform are as follows:

$$
\begin{aligned}
s_{sub-LFM-BPSK}(t) \\
= \sum_{i=1}^{N} b_i \times u(t - iT_b) \times \exp(j \times \pi \times u \times t^2 + j \times \theta_i) \\
= \sum_{i=1}^{N} u(t - iT_b) \times \exp(j \times \pi \times u \times t^2 + j \times b_i \times \theta_i)
\end{aligned}
\tag{3}
$$

In Equation (3), $u(t) = \begin{cases} 1, & 0 < t < T_b \\ 0, & elsewhere \end{cases}$, the up$-$LFM duration is $T$, the bandwidth is $B$, and an LFM pulse can carry $N$ bits; each bit duration is $T_b = T/N$, $b_i$ is the bit value, $\theta_i$ is the phase shift, and $u = B/T$ is frequency modulation. Using the spread spectrum processing of Equation (3), integrated omnidirectional waveform Equation (4) is generated:

$$
s_{\text{JRC}}(t) = \sum_{i=1}^{N} \sum_{k=1}^{M} u_c(t - iT_b - kT_c) \times \exp(j \cdot (\pi \times u \times t^2 + b_i \times c_k \times \theta_i))
\tag{4}
$$

where $T_c$ represents the duration of the spectrum code, and $c_k$ represents the value of the spread spectrum code. Due to the spread spectrum processing, Equation (4) is equivalent to the microbit processing of Equation (3).

The radar performance and communication performance of Equations (3) and (4) have been analyzed in detail in the literature [21]. Due to the integrated design based on an LFM waveform, the integrated waveform can carry several bits of communication data in one section of the LFM signal, the radar detection performance loss is limited, and it has excellent range and velocity resolution performance. From the communication point of view, due to the spread spectrum code, the single-antenna integrated waveform can be considered as communication data with reasonable spread spectrum processing. In the MIMO UAV scenario, each transmitting antenna can carry communication data after different spread spectrum code processing, and it has orthogonal characteristics, which can ensure that the waveform of each antenna has orthogonality. From the perspective of analytical convenience, Equation (3) can be rewritten as follows:

$$
s_{sub-LFM-BPSK}(t) = \exp(j \times \pi \times u \times t^2 + \theta_{data}(t))
\tag{5}
$$

In Equation (5), $\theta_{data}(t) = \begin{cases} \theta(t - iT_b) \in [-\theta, \theta] \\ 0 \end{cases}$, where $\theta$ represents a series of specific communication values. When the $\theta$ value range is $\pi/2$, it represents BPSK communication data. $i = 0, 1 \ldots N - 1$ represents the communication data carried by the LFM radar waveform, and the quantity is $N$. $N \cdot T_b$ is the time width of the integrated waveform pulse, and the dimension of (5) in the time domain is $1 \times N_s$. $N_s$ is the sampling frequency of the integrated waveform.

Considering the factor of spread spectrum code, (4) can be rewritten as follows:

$$
s_{\text{JRC}}(t) = \exp(j \times \pi \times u \times t^2 + \theta_{data}(t)v(t))
\tag{6}
$$

In the integrated waveform expression (6), $v(t)$ represents the bit sequence of the spread spectrum code. The process of spread spectrum is equivalent to the micro-bit processing of the communication data in Equation (5).

### 2.2. General Expression of Integrated Omnidirectional Waveform

In Section 2.1, an integrated omnidirectional waveform expression of the UAV scenario is constructed using the orthogonal spread spectrum code. This section considers the representation of the closed-form expressions of omnidirectional radar communication integrated orthogonal waveform under certain communication performance constraints. In

the ideal model, the general orthogonal expression of the UAV integrated system waveform can be derived assuming that the communication receiver knows the communication constellation distribution in the specific integrated waveform from the communication perspective, and the UAV system has predicted the communication channel characteristics. It is assumed that the number of transmitting antennas in the UAV radar communication integration system is $N_t$, and the target omnidirectional detection is carried out while serving $Q$ downlink single-antenna users. The number of data samples of downlink communication users is $K$; thus, the received signal matrix of a single downlink communication user can be expressed as follows:

$$Y_c = H_{JTC}X + W \tag{7}$$

In Equation (7), $X = [x_1, x_2, \ldots x_K] \in \mathbb{C}^{N_t \times K}$ is the integrated waveform matrix. Matrix $H_{JTC} = [h_1, h_2, \ldots h_Q] \in \mathbb{C}^{Q \times N_t}$ is the channel matrix between the UAV and the communication user, and the channel characteristics can be accurately estimated. Matrix $W = [w_1, w_2, \ldots w_K] \in \mathbb{C}^{Q \times K}$ is the noise matrix. Assuming that matrix $C = [c_1, c_2, \ldots c_Q] \in \mathbb{C}^{Q \times K}$ is the communication symbol matrix sent by the integrated transmitter to a single-antenna user, Equation (7) can be rewritten as follows:

$$Y_c = C + (H_{JTC}X - C) + W \tag{8}$$

In Equation (8), for a single downlink communication user, $(H_{JTC}X - C)$ refers to the multi-user interference of a single downlink communication user, which is directly related to the reachability and rate of the entire system. The smaller the value $(H_{JTC}X - C)$, the greater the communication reachability of the integrated system. Under the condition of a fixed geometric distribution of the MIMO antenna array, the transmitted beam pattern of the integrated terminal is determined by the waveform covariance matrix and is limited by the total power of the transmitting end. In this case, the omnidirectional integrated waveform must be orthogonal; i.e., the corresponding covariance matrix is the unit matrix. The design of the omnidirectional integrated waveform of a UAV can be formulated as the following optimization problem provided that the minimum interference of multiple users is ensured:

$$\min_X \|H_{JTC}X - C\|_F^2$$
$$s.t. \ \frac{XX^H}{K} = \frac{P_T}{N_t} I_{N_t} \tag{9}$$

In Equation (9), $P_T$ is the total transmitted power of the integrated terminal UAV. The constraints ensure that the radar beam is omnidirectional, which is referred to as the strict omnidirectional waveform. The derivation of the global optimal solution of the waveform can be found in Appendix A. The final closed-form expression is as follows:

$$X = \sqrt{\frac{KP_T}{N_t}} U I_{N_t \times K} V^H \tag{10}$$

In Equation (10), $U \sum V^H = H_{JTC}^H C$ is the singular value decomposition of matrix $H_{JTC}^H C$. The integrated orthogonal waveform can be designed under the condition of predicting communication symbol $C$'s constellation and channel state information $H_{JTC}$ of the integrated waveform and ensuring the communication performance. At this point, the UAV can construct integrated signals to realize omnidirectional detection and communication functions at the same time.

## 3. Performance Analysis of the UAV MIMO Orthogonal Waveform

In the UAV MIMO antenna scenario, the above expression (6) represents the monopulse transmit waveform of a single antenna. From the perspective of communication, the integrated transmission system of a UAV has $N_t$ different antennas, and different antennas adopt different spread spectrum codes; thus, the UAV can realize the orthogonal waveform

between different antenna waveforms, which can satisfy the waveform requirements of MIMO radar detection. At this point, the key to the construction of an omnidirectional detection waveform in a UAV integrated system lies in the selection of a large number of spread spectrum codes with superior performance and the spread spectrum processing for the transmitted data of each antenna to realize the construction of an orthogonal waveform. In study [21], the performance of various spread spectrum codes was analyzed, and as a result, a type of complex chaotic spread spectrum code with superior performance and large numbers was proposed for use. Spread spectrum codes can construct the orthogonal waveform by using a chaotic spread spectrum and modifying the initial parameters. The UAV MIMO radar communication integration system realizes not only radar detection but also the communication function in the specific application of the orthogonal waveform. To improve the communication throughput, it is assumed that UAV MIMO antennas transmit different communication data and different spread spectrum codes to realize the orthogonal waveform. In this application scenario, the final transmission waveform of the UAV MIMO radar communication integration system is as follows:

$$x_{\text{MIMO-JRC}}(t) = \sum_{n=1}^{N_t} \exp(j \times \pi \times u \times t^2 + \theta_{data\_n}(t)v_n(t)) \tag{11}$$

The iintegrated signal can be considered a special communication signal for which its carrier is replaced by the LFM signal instead of an ordinary fixed high-frequency signal. From the point of view of MIMO communication, each antenna of the transmitting antenna array can transmit different communication symbols at the same time. If one communication symbol is sent by different antennas at the same time as one frame, the communication data of one frame can be expressed as follows:

$$\theta_{data\_f}(t) = [\theta_{data\_1}(t)v_1(t) \ \ \theta_{data\_2}(t)v_2(t) \ \dots \ \theta_{data\_N_t}(t)v_n(t)]^T \tag{12}$$

The dimension of the time domain is $N_t \times N_s$. In the time domain, the integrated signal of each frame can be expressed as follows:

$$x_{JRC\_f}(t) = [x_{JRC\_1}(t) \ \ x_{JRC\_2}(t) \ \dots \ x_{JRC\_N_t}(t)]^T \tag{13}$$

The transmission waveform of the UAV integrated system simultaneously performs the detection function and communication function. When targets are detected, the integrated waveform generates an echo signal, which is received by the radar's receiving array in the integrated UAV. At the same time, the communication receiver at the far field receives the integrated waveform signal and obtains the communication signal after a series of demodulation processing. When the integrated system detects the *L* target and generates echoes, the echo signals received by the receiving array can be expressed as follows:

$$x_r(t) = \varepsilon_l \sum_{l=1}^{L} b_l^c(\varphi_l) \sum_{n=1}^{N_t} a_n^H(\varphi_l) \exp(j \times \pi \times u \times (t - t_l)^2 + \phi_{data\_n}(t - t_l)v_n(t - t_l)) \\ + w(t) \tag{14}$$

In Equation (14), $\varepsilon_l$ is the reflection coefficient of the target, which is related to the RCS of the *L* targets; $t_l$ is the time delay of the echo; $\varphi_l$ is the azimuth of the target. The distance information of the target can be obtained by using a matched filter to compress the echo pulse at the receiving end. The echo waveform signal is matched to the filter. The matching equation is as follows:

$$y_r(t) = \int_0^{N \cdot T_b} x_{MIMO-JRC}^c(t) x_r(t) d\tau \tag{15}$$

The time delay (distance) information of the target in the corresponding azimuth can be extracted by matching filtering, and the azimuth–range graph can be obtained to display the target's distance information.

It is assumed that the communication user is located in the far field. The distance and azimuth of the entire column of the integrated antenna can be predicted. For the integrated waveform of the MIMO transmitting array, the signal received by the communication receiver is as follows:

$$x_C(t) = \sum_{n=1}^{N_t} a_n(\theta_c) \exp(j \times \pi \times u \times (t - t_c)^2 + \theta_{data\_n}(t - t_c)v_n(t - t_c)) + w(t) \qquad (16)$$

In Equation (16), $t_c$ is the signal delay of the integrated signal reaching the communication terminal. According to the above assumptions, the communication receiving terminal can be estimated in advance. The azimuth of the integrated sending end is relative to the communication receiving end, and $w(t)$ is the noise signal. When the communication receiving terminal extracts data information, the expression of the extracted signal is as follows:

$$x_{refc}(t) = \sum_{n=1}^{N_t} a_n^H(\theta_c) \exp(j \times \pi \times u \times t^2 + v_n(t)) \qquad (17)$$

The eextracted signal is multiplied by the signal of the communication receiver to obtain the communication signal:

$$y_c(t) = \int_0^{N \cdot T_b} x_{refc}^c(t) x_C(t) d\tau \qquad (18)$$

The above analysis process analyzes the single-antenna user's communication data extraction method. The communication data transmitted by the UAV's integrated system can be obtained via appropriate decoding, demodulation and spread spectrum coding methods. At present, each UAV antenna transmits different data with a different spread spectrum code, and the communication receiver uses the same demodulation mode to demodulate the communication data of each antenna, which can greatly increase the communication throughput, and its incremental performance is related to the number of antennas. In the multi-user scenario, the performance analysis is equivalent to the simple overlay of a single user, and the analysis process can be ignored.

The radar communication integration technology based on the MIMO antenna can use the orthogonal waveform set to form a wide beam to continuously detect all directions and simultaneously realize omnidirectional communication with better space and time efficiency. Therefore, the design of an integrated orthogonal waveform set is a key technology for MIMO antennas. Sub-LFM-BPSK is a relatively superior single-antenna integrated waveform. It introduces a spread spectrum code sequence for the spread spectrum processing of the waveform of each MIMO transmit antenna. Firstly, the orthogonal design of each antenna is implemented so that the waveform set of each antenna in the integrated system is suitable for omnidirectional orthogonal MIMO antenna scenarios. At the same time, after spread spectrum processing, the communication performance is compensated, and the communication reliability is improved. In this paper, the integrated orthogonal waveform set can realize radar detection and communication at the same time, which not only retains the excellent detection performance of LFM but also has excellent communication performance.

## 4. Simulation

In this section, we have provided computer simulations based on MATLAB. The detection performance of the omnidirectional integrated waveform of UAV MIMO is simulated and tested. The simulation parameter settings are listed in Table 2.

**Table 2.** Omnidirectional integrated waveform simulation parameters of the UAV.

| Parameter | Value |
|---|---|
| Time width $T$ | 10 µs |
| Bandwidth $B$ | 50 MHz |
| Sampling rate $f_s$ | 500 MHz |
| Frequency modulated $u$ | $5 \times 10^{11}$ |
| Spread code $M$ | $15-$bit chaotic code |
| Communication data $N$ | 50 |
| Phase shift $\phi_k$ | $\pi/6$ |
| Data width $T_b$ | $T/N$ |
| Spread code width $T_c$ | $T_b/M$ |
| Transmit antennas $N_t$ | 4 |
| Receive antennas $N_r$ | 4 |
| Antenna space $d$ | $\lambda/2$ |
| Target azimuth | $120°$ |
| PRF | 1 kHz |

Considering the limitations such as the volume of the UAV, it is assumed that the number of transceiver antennas in the UAV MIMO integrated system is four. The communication modulation is sub$-$LFM$-$BPSK, and the phase shift between binary bit "1" and "0" becomes $\pi/6$ and $-\pi/6$. After the transmission waveforms are processed by the spread spectrum code, which uses a 15-bit chaotic spread spectrum code, the omnidirectional orthogonal waveforms are generated.

A single LFM radar pulse signal is embedded with 50 bits of communication data, the PRF is 1 KHz, and the total communication rate of the integrated MIMO system is 200 KHz/s. The overall system's communication rate is varied by changing the time width of the communication code, the number of integrated system antenna arrays, and the integrated PRF waveform. The integrated system transmits omnidirectional integrated waveforms for target detection and communication. It is assumed that the detected target is a single-point target located 350 m from the integrated UAV and that the azimuth angle is $120°$. To improve the overall communication throughput of the system, different antennas of the transmit antenna array transmit different communication data.

The ambiguity function [22] is a commonly used performance index for common radar communication signals, and it is an important measure of radar performance that reflects the radar's detection capability. The expression of the ambiguity function of the integrated signal $x(t)$ is $\chi(\tau, f_d) = \left| \int_0^T x(t) \cdot x^c(t-\tau) \cdot e^{j \cdot 2\pi \cdot f_d \cdot t} dt \right|$, where $\tau$ is the time delay, and $f_d$ is the Doppler frequency shift. The initial LFM waveform, the LFM$-$BPSK wave, the sub$-$LFM$-$BPSK waveform, and the 15-bit spread spectrum orthogonal waveform are compared as follows.

An ideal radar signal's ambiguity function is in the shape of a thumbtack; the top view of the ambiguity function is a point on the plane. The ideal radar signal does not exist in practice; however, as shown in Figure 2a, the LFM signal is an excellent radar signal: Its ambiguity function has an oblique blade shape, and the top view of the ambiguity function is an oblique line. As shown in Figure 2b, after adding data, the ambiguity function diffuses, which affects the radar detection performance. From an analysis of the literature [21], it can be seen that the phase shift of the communication causes the diffusion of the ambiguity function. As the phase shift decreases, the diffusion effect decreases. Therefore, as shown in Figure 2c, the sub$-$LFM$-$BPSK modulation method with a small phase shift is adopted to improve the radar's performance. We expect the integrated signal designed based on the LFM radar waveform to have ambiguity function characteristics similar to those of the LFM waveform, and we expect it to preserve the radar's characteristics while realizing communication as much as possible. The single-antenna waveform with spread spectrum code processing has excellent ambiguity function characteristics when used in a MIMO system. As shown in Figure 2d, after the spread spectrum processing of the integrated

signal, the top view of its ambiguity function is similar to Figure 2a. At the same time, the MIMO system realizes the orthogonal waveform of each antenna and constructs the omnidirectional waveform of MIMO radar communication integration.

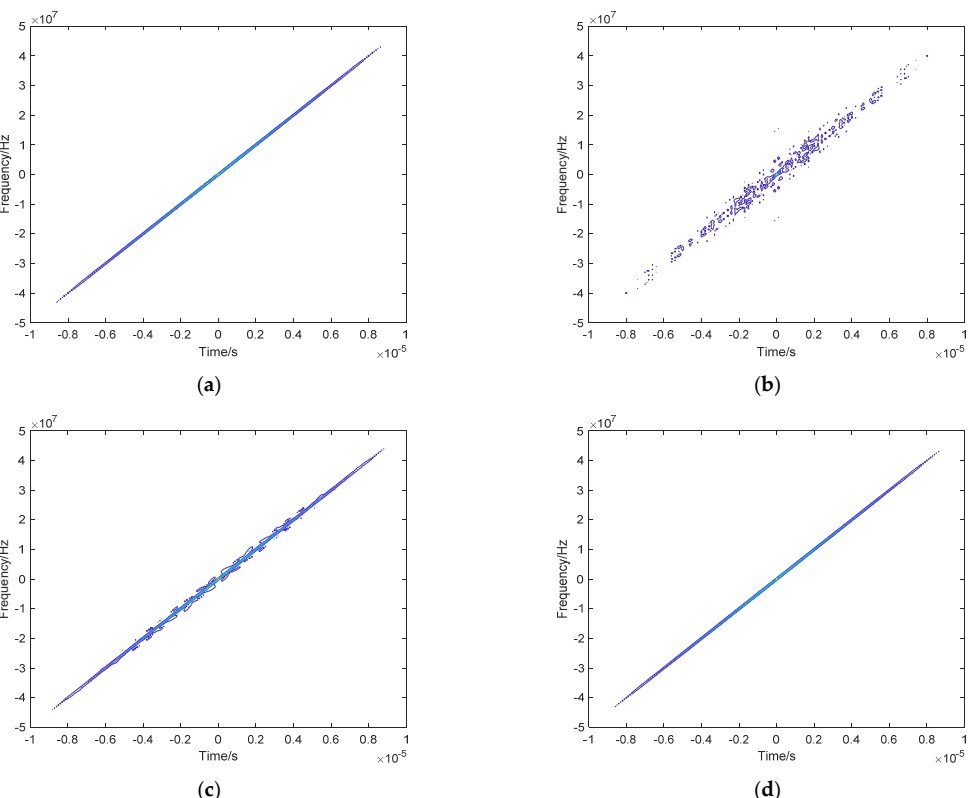

**Figure 2.** Top view of the ambiguity function: (**a**) LFM waveform; (**b**) LFM−BPSK waveform; (**c**) sub−LFM−BPSK waveform; (**d**) 15−bit spread spectrum orthogonal waveform.

As shown in Figure 3, when the LFM radar waveform is used with MIMO antennas, there is no loss of radar detection performance as it does not carry communication data. The low side lobes of the echo in the pulse compression diagram in Figure 3a indicate that the LFM waveform has good radar detection performance. As shown in Figure 3a, we can clearly distinguish the target's position and angle.

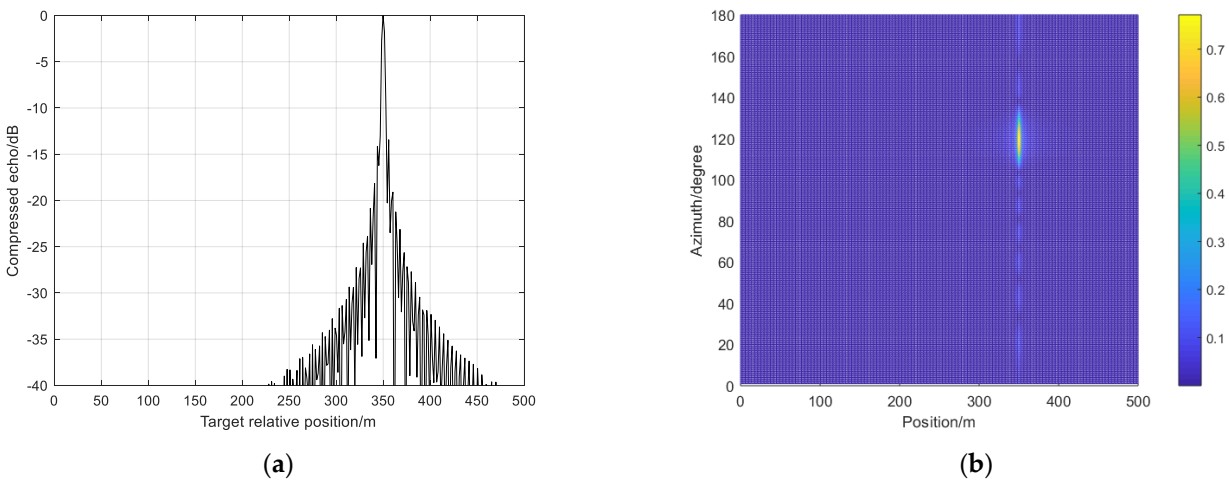

**Figure 3.** LFM signal: (**a**) echo pulse compression diagram; (**b**) echo azimuth–range diagram.

As shown in Figure 4, since communication data are carried in the LFM-BPSK integrated waveform, although the communication function is achieved, the introduction of

communication data in the integrated waveform results in a decrease in range performance. As shown in Figure 4a, the pulse compression signal of the echo has large fluctuations in its side lobes, and its range performance is lost. As shown in Figure 4b, we cannot distinguish the position and angle of the target.

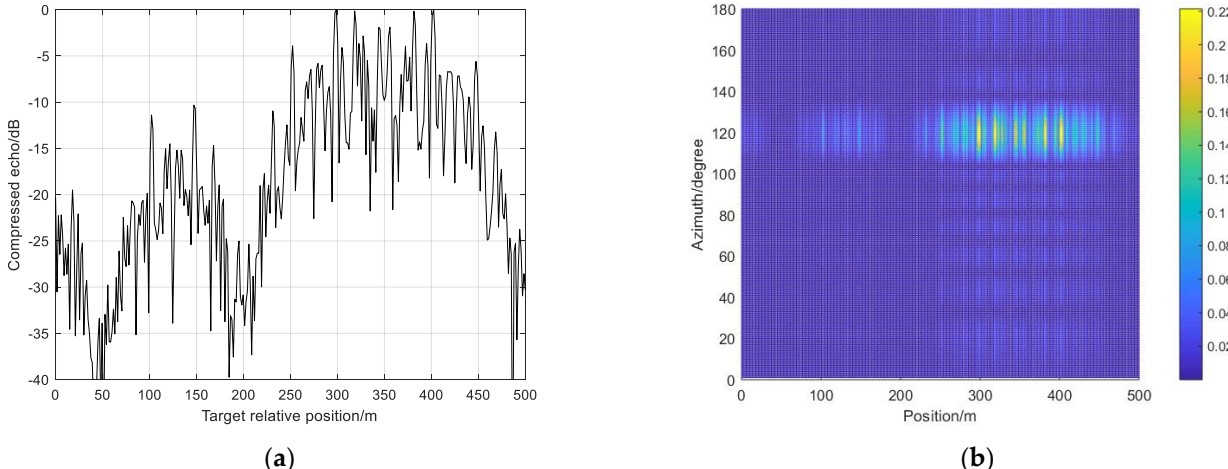

(**a**)     (**b**)

**Figure 4.** LFM−BPSK radar communication signal: (**a**) echo pulse compression diagram; (**b**) echo azimuth–range diagram.

As shown in Figure 5, the range performance of the integrated sub−LFM−BPSK waveform was significantly improved compared with that in Figure 4, and as shown in Figure 5a, the pulse compression signal of the echo was reduced in its side lobes. As shown in Figure 5b, we can discriminate the position and angle of the target.

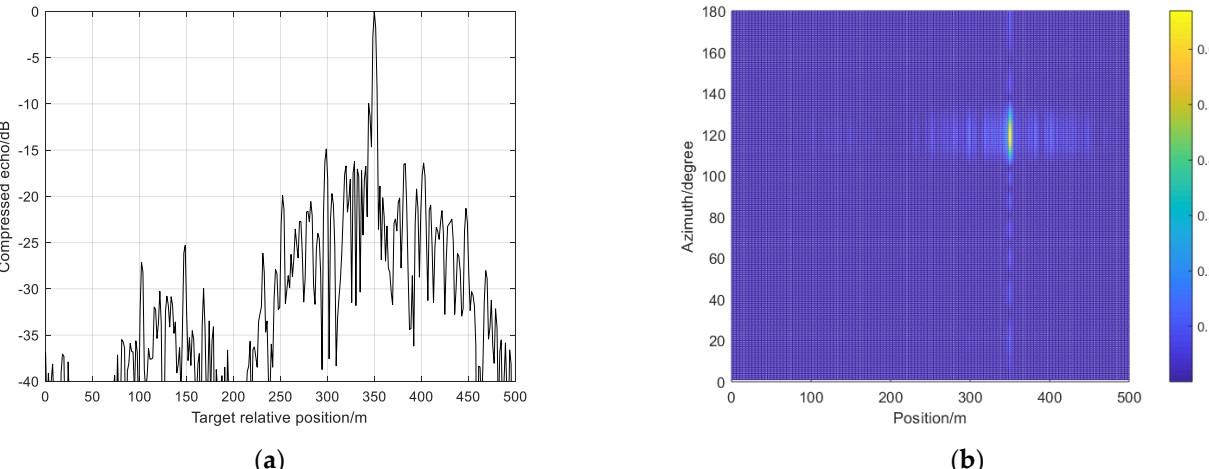

(**a**)     (**b**)

**Figure 5.** Sub−LFM−BPSK radar communication signal: (**a**) echo pulse compression diagram; (**b**) echo azimuth–range diagram.

As shown in Figure 6, when the spread spectrum code is used, the side-lobe level of the echo pulse compression curve gradually decreases, but theoretically, it cannot be eliminated due to the presence of communication interference. The characteristics shown in Figures 5 and 6 are mutually consistent with the ambiguity function characteristics in Figure 2, indicating that the integrated waveform after the incorporation of the orthogonal spread spectrum has excellent radar characteristics. The range characteristics and azimuth–range graphs of several waveforms are analyzed in Figures 3–6. The bit error rate and spectral performance can be used to analyze the communication performance of the integrated waveform. It can be observed in study [21] that the bit error rate of the

integrated waveform designed in this paper is equivalent to that of sub−BPSK. The bit error rate discussion is ignored in this paper.

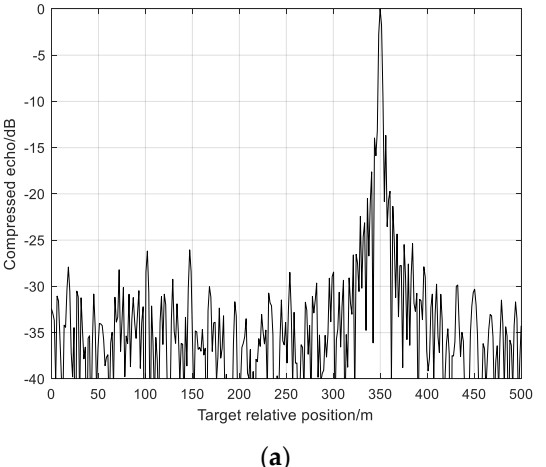

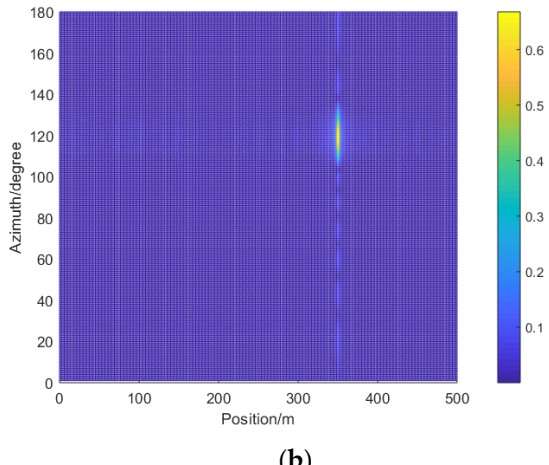

(**a**)                                                                                    (**b**)

**Figure 6.** The 15−bit spread spectrum orthogonal signal: (**a**) echo pulse compression diagram; (**b**) echo azimuth–range diagram.

As shown in Figure 7, the spectrums of several waveforms are compared. When compared with the original LFM waveform, spectrum broadening and spectrum amplitude oscillation inevitably occur after LFM is embedded in communication data. After phase shift reduction and spread spectrum processing, the spectrum expansion and amplitude oscillation of the integrated waveform will be reduced, and the communication performance of the omnidirectional integrated waveform is improved.

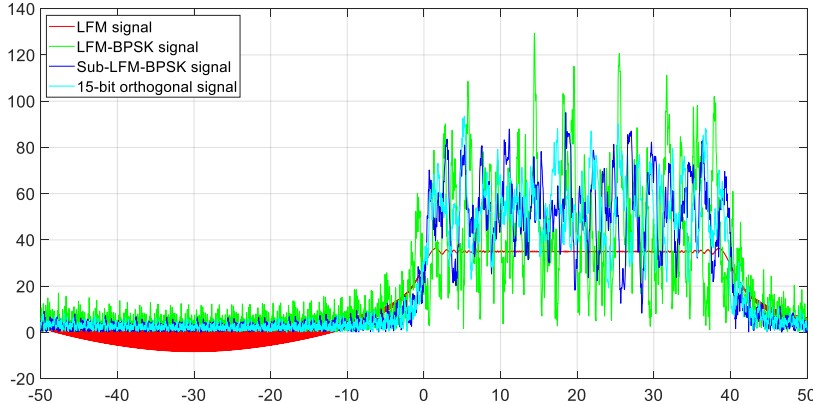

**Figure 7.** Spectrum comparison diagram.

When a single-antenna integrated signal communicates, the communication rate is affected by the embedded number of communication codes and the pulse repetition rate. In the MIMO antenna radar communication integration system, the reception rate of its communication receiver greatly improved due to the omnidirectional orthogonal waveform transmission and the use of multiple antennas sending different data simultaneously.

## 5. Conclusions

Orthogonal waveform design is a challenging task in the field of MIMO radar communication integration. This paper proposes an orthogonal waveform design method that can be used in the UAV MIMO scenario based on the single-antenna integration scheme; the paper derives the closed-form expressions of the radar communication integrated orthogonal waveform under certain communication performance constraints. By deriving the orthogonal waveform expression, the ambiguity function, echo suppression

performance and communication spectrum performance are simulated and analyzed. The simulation results show that the UAV MIMO orthogonal waveform method proposed in this paper has excellent range and communication performance. The orthogonal waveform makes full use of the technical advantages of the LFM radar waveform and MIMO multi-antenna and considers the application scenarios of UAVs to achieve radar detection and communication functions.

**Author Contributions:** Conceptualization and methodology, H.M., J.W. and X.S.; software, validation and formal analysis, H.M. and J.W.; writing—original draft preparation, H.M.; writing—review and editing, H.M., J.W., W.J. and X.S.; funding acquisition, X.S. All authors have read and agreed to the published version of the manuscript.

**Funding:** This research was funded by Fundamental Research Funds for the Central Universities 2021RC260.

**Data Availability Statement:** The codes used in this manuscript are available from the corresponding author upon request.

**Acknowledgments:** We would like to express our gratitude to the academic editor and anonymous reviewers for their constructive comments and suggestions.

**Conflicts of Interest:** The authors declare no conflict of interest.

## Appendix A

Optimization problem (9) can be rewritten as follows:

$$
\begin{aligned}
\|H_{JTC}X - C\|_F &= tr\left( \left( H_{JTC}X - C \right) \left( H_{JTC}X - C \right)^H \right) \\
&= tr\left( H_{JTC}XX^H H_{JTC}^H \right) - 2\mathrm{Re}(tr(CX^H H_{JTC}^H)) + tr(CC^H) \\
&= \frac{KP_t}{N} tr\left( H_{JTC}H_{JTC}^H \right) - 2\mathrm{Re}(tr(CX^H H_{JTC}^H)) + tr(CC^H)
\end{aligned}
\tag{A1}
$$

Then, th optimization minimization problem is equivalent to the maximization of $\mathrm{Re}(tr(X^H H_{JTC}^H C))$, assuming that the singular value of the matrix $H_{JTC}^H C$ is decomposed into $U \sum V^H$, and $D$ is defined as $D = V^H X^H U$; $\sum \in \mathbb{C}^{N_t \times K}$ is a diagonal matrix, considering the cyclic invariant property of the matrix trace, and $\mathrm{Re}(tr(X^H H_{JTC}^H C))$ can be rewritten as follows:

$$
\begin{aligned}
\mathrm{Re}(tr(X^H H_{JTC}^H C)) &= \mathrm{Re}(tr(X^H U \sum V^H)) \\
&= \mathrm{Re}(tr(V^H X^H U \sum)) \\
&= \mathrm{Re}(tr(D \sum)) \\
&= \mathrm{Re}(tr(\sum_{i=1}^{N} D_{ii} \sum_{ii}))
\end{aligned}
\tag{A2}
$$

Therefore for maximization (A2), $D$ can be expressed as follows:

$$
D = V^H X^H U = \sqrt{\frac{KP_t}{N_t}} I_{N_t \times K}
\tag{A3}
$$

Then, the global optimal solution of the original problem can be expressed as follows:

$$
X = \sqrt{\frac{KP_T}{N_t}} U I_{N_t \times K} V^H
\tag{A4}
$$

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
