# Peer review of "Integrated Waveform Design Based on UAV MIMO Joint Radar Communication"

_information, doi:10.3390/info14080455_

Round 1

Reviewer 1 Report

 1. In the Unmanned Aerial Vehicle (UAV) MIMO antenna scenario, orthogonal integrated waveform suitable for MIMO antenna is proposed by using sub LFM-BPSK waveform combined with chaotic spread spectrum code.

2. In the figure 2, top view of ambiguity function: (a) LFM waveform; (b) sub LFM - BPSK waveform ; (c) 15-bit spread spectrum orthogonal waveform; (d) 31-bit spread spectrum orthogonal wave-form, should be elaborated in detail.

3. In the figure 4, integrated echo signal of sub LFM-BPSK radar communication: (a) time domain diagram of single antenna; (b) echo pulse compression diagram; (c) echo azimuth diagram; (d) echo azimuth - range diagram, should be elaborated in detail.

4.  Please elaborate the technical features and technique efficacy of orthogonal integrated waveform suitable for MIMO antenna in detail.

5.Please compare the contributions of the proposed technology to related technologies, in detail. Consider adding a detailed discussion concerning related technologies.

6.Please discuss some of the future applications of the proposed technology, in detail to hold the reader’s interest.

7.Please thoroughly revise the language before your final submission.

Moderate editing of English language required.

Reviewer 2 Report

Authors investigated the integrated orthogonal waveform set in MIMO scenario is constructed by adopting the spreading code in orthogonal method. However, before acceptance, major corrections are required:

1.       What is the meaning of LFM-BPSK, LFM-MSK, and LFM-CPM?

2.       Please improve the figure 1 quality.

3.       Please check the eqs. (1) and (2). And also give the reference no.

4.       Typo error (, after eq. (3).

5.       Typo error i, after eq. (7).

6.       Which software is used for running the simulation like Python or Matlab.

7.       Results section is NOT clear. Please improve the all figures quality in simulation section. Please compare with previous results.

8.       It is recommended to check the full paper by native English speaker.

 It is recommended to check the full paper by native English speaker.

Round 2

Reviewer 2 Report

Thanks for correction.

Good.